# High magnesium mobility in ternary spinel chalcogenides

Pieremanuele Canepa [1,2], Shou-Hang Bo[1,2,5], Gopalakrishnan Sai Gautam [1,2,3], Baris Key[4], William D. Richards[2], Tan Shi [3], Yaosen Tian[3], Yan Wang[2], Juchuan Li [1] & Gerbrand Ceder[1,2,3]

Magnesium batteries appear a viable alternative to overcome the safety and energy density limitations faced by current lithium-ion technology. The development of a competitive magnesium battery is plagued by the existing notion of poor magnesium mobility in solids. Here we demonstrate by using ab initio calculations, nuclear magnetic resonance, and impedance spectroscopy measurements that substantial magnesium ion mobility can indeed be achieved in close-packed frameworks (~ 0.01–0.1 mS cm$^{-1}$ at 298 K), specifically in the magnesium scandium selenide spinel. Our theoretical predictions also indicate that high magnesium ion mobility is possible in other chalcogenide spinels, opening the door for the realization of other magnesium solid ionic conductors and the eventual development of an all-solid-state magnesium battery.

[1] Materials Science Division, Lawrence Berkeley National Laboratory, Berkeley, CA 94720, USA. [2] Department of Materials Science and Engineering, Massachusetts Institute of Technology, Cambridge, MA 02139, USA. [3] Department of Materials Science and Engineering, University of California Berkeley, Berkeley, CA 94720, USA. [4] Chemical Sciences and Engineering Division, Argonne National Laboratory, Argonne, IL 60439, USA. [5]Present address: University of Michigan—Shanghai Jiao Tong University Joint Institute, Shanghai Jiao Tong University, 800 Dong Chuan Road, Minhang District, Shanghai 200240, China. Pieremanuele Canepa and Shou-Hang Bo contributed equally to this work. Correspondence and requests for materials should be addressed to University.of.Michigan—Shanghai.Jiao.Tong.University.Joint.Institute, Shanghai.Jiao.Tong.University, 800 Dong.Chuan.Road, Minhang.District, Shanghai 200240ChinaP.C. (email: pcanepa@lbl.gov) or to S.-H.B. (email: shouhang.bo@sjtu.edu.cn) or to G.C. (email: gceder@berkeley.edu)

Developing new battery technologies to sustain the ever-growing demand of energy storage constitutes one of the greatest scientific and societal challenges of the century. Lithium-ion batteries (LIBs) are at the center of this energy revolution: they power millions of portable electronics, electric vehicles, and are even seeing introduction into the electric grid. Li ion's success is in part due to the remarkable mobility of $Li^+$ in many solids. Fast Li-ion transport enables intercalation electrodes, in which charge is stored by moving the ions in and out of crystal structures. More recently, super-ionic conductivity of $Li^+$ in solids, greater than 1 mS cm$^{-1}$, has instigated a renewed interest in solid-state LIBs[1, 2], which would have substantial advantages in terms of safety and lifetime.

A technology that has the potential to alleviate resource issues with Li-ion systems and further increase the energy density is $Mg^{2+}$ intercalation systems[3, 4]. Replacing Li with safer and earth-abundant Mg[3, 5, 6], has the advantage of doubling the total charge per ion, resulting in larger theoretical volumetric capacity compared with typical LIB. Most importantly, in Mg batteries (MB) the anode is constituted by energy dense Mg metal (~ 3,830 Ah l$^{-1}$) notably surpassing the theoretical volumetric energy density of the current graphitic anode of LIB (~ 700 Ah l$^{-1}$) and even that of lithium metal (2,062 Ah l$^{-1}$)[5,6].

A generally perceived obstacle to the development of Mg-battery technology is the low mobility of $Mg^{2+}$ in solids[7, 8]. Indeed, poor mobility of $Mg^{2+}$ (and other multivalent cations[9]) prevents the development of a broad spectrum of cathode materials, as are available to LIB and Na-ion battery technologies. Poor Mg transport also limits the use of solid barrier coatings to protect electrodes from reaction with the liquid electrolyte, or the development of full solid-state MBs, which would alleviate many of today's problems caused by liquid electrolytes. Pioneering experimental[10–17] and theoretical[14, 18] studies, aiming to develop solid and semi-solid multi-valent high-temperature conductors, have reported good conductivity at elevated temperatures (~3 × 10$^{-2}$–10 mS cm$^{-1}$, 400–800 °C) but high room temperature conductivity has remained elusive, demonstrating the challenges posed by the slow migration of $Mg^{2+}$ ions.

In this study we show that high $Mg^{2+}$ mobility in solids can be achieved by judicious tuning of crystal structure and chemistry. By combining ab initio calculations, synchrotron X-ray diffraction (XRD), electrochemical impedance spectroscopy and solid-state nuclear magnetic resonance (SS-NMR), we demonstrate facile $Mg^{2+}$ conduction at room temperature. Experimentally, we demonstrate the discovery of the first generation of crystalline solids, i.e., spinel $MgX_2Z_4$, with X = (In, Y, Sc) and Z = (S, Se), which possess high $Mg^{2+}$ cation mobility at room temperature. In addition, we propose practical design rules to identify fast multivalent-ion solid conductors. Our theoretical calculations and electrochemical experiments suggest that sulfide and selenide spinels can potentially integrate with current state-of-the-art Mg cathodes, e.g., spinel-$MgTi_2S_4$ and Chevrel-$Mo_6S_8$[3, 19].

## Results

**Computational screening of magnesium solid electrolytes.** A variety of structural and electronic design principles for fast cation mobility in materials have been proposed[20, 21]. For example, it has been shown that Mg migration can be facilitated by propping open layered structures, such as $MoS_2$ and $V_2O_5$[22, 23]. It is recognized from a statistical analysis of the Inorganic Crystal Structural Database[24] that $Mg^{2+}$ highly favors octahedral coordination environment in oxides and sulfides[25, 21], whereas $Zn^{2+}$ prefers tetrahedral coordination. In parallel, recent computational studies[25, 21] have indicated that fast motion of an ion can be achieved when the stable site for the ion has an

unfavorable coordination and the activated state has a more favorable coordination. The decrease of the activated state energy and increase of the initial and final stable state energies leads to a flattening of the energy profile along the migration path and hence fast ion diffusion. Consequently, structural frameworks with high Mg (or Zn) mobility must display Mg (Zn) residing in its unfavorable anion coordination environment, i.e., ≠ 6 (≠ 4 for $Zn^{2+}$)[20, 21, 25, 26]. Indeed, when Mg or Zn occupy a tetrahedral site—preferred by Zn but not by Mg—in the same structural framework, the migration barriers for $Zn^{2+}$ ions can be twice as large as that of $Mg^{2+}$[25, 21]. In contrast, properly tailoring the structure to the coordination preference of the $Zn^{2+}$ ion[27] can lead to high rate $Zn^{2+}$ intercalation batteries[9, 28].

Minimizing the change in coordination environment along the migration path further contributes to a lower barrier, as it keeps the energy landscape flat[20, 21]. This principle led to the identification of body centered cubic (BCC) anion packing as the best structural motif for Li-ion conductors[20]. Although one would similarly expect such BCC frameworks to have good Mg mobility, no BCC-packed anion structures with Mg are known. Hence, the search for good Mg conductors is confined to close-packed frameworks in which Mg does not reside in octahedral anion coordination. Spinel structures are therefore expected to be reasonable $Mg^{2+}$ conductors as the electrostatics of the cation arrangement makes the stable site the tetrahedral (tet) site[24]. Indeed, Yin et al.[29] have reversibly intercalated Mg into spinel-$MgMn_2O_4$ from an organic electrolyte, whereas Kim et al.[30] employed complex $^{25}Mg$ NMR measurements to verify Mg intercalation in spinel-$Mn_2O_4$ using aqueous electrolytes.

Migration barriers in a structure can be further reduced if the volume per anion is increased, with the magnitude of the barrier closely following the order $O^{2-} > S^{2-} > Se^{2-} > Te^{2-}$[20, 31]. An increase in the volume of these divalent anions also results in increased electric polarizability (the ability to deform the anion electronic charge density by the charge of a nearby cation), which in turn influences cation mobility. Thus, maximizing the volume per anion of the structure and its electric polarizability constitute another design criterion for multivalent ion conductors. Indeed, a recent theoretical report of sulfide spinels suggested that some of them may function as Mg-insertion cathodes[32], with the spinel-$Mg_xTi_2S_4$ demonstrated to work experimentally by Sun et al.[19], albeit at elevated temperature.

The analysis above (discussed in more detail in Supplementary Notes 1 and 9, and Supplementary Tables 5, 6 and 7) leads us to investigate sulfide and selenide spinels with stoichiometry $MgX_2Z_4$ (where Z = S and Se and X = In, Y and Sc)[33, 34]. In spinel structures, the ion migration between two tetrahedral sites (tet) occurs via a vacant octahedral site (oct), which face-shares with the tetrahedral sites, following the migration topology tet–oct–tet of Fig. 1a, b. The magnitude of the migration barrier is determined by the energy of the migrating ion in the shared triangular face between oct and tet sites ($E_a$, Fig. 1b), which in turn is influenced by the size of that triangular face, and by the anion species that form this triangle (Fig. 1b).

Mg and Zn migration barriers (orange bars) obtained with first principles (density functional theory, DFT)-based nudged elastic band (NEB)[35] calculations in spinel $AX_2Z_4$ structures (with A = Mg or Zn, X = Sc, Y and In, and Z = S, Se and Te) are shown in Fig. 1c, d vs. the volume per anion (blue bars). Extrapolated from ab initio molecular dynamic simulations (AIMDs), Fig. 1e, f display the Mg probability density in $MgSc_2Se_4$ and the trend of diffusivities as a function of temperature in $MgSc_2Se_4$ and $MgY_2Se_4$, respectively.

Of the chalcogenides in Fig. 1, only $MgSc_2S_4$[33], $MgIn_2S_4$[33, 36], $MgSc_2Se_4$[34], $MgY_2Se_4$[34], $ZnSc_2S_4$[36], $ZnY_2S_4$[36], $ZnIn_2S_4$[37], and $ZnY_2Se_4$[38] have been experimentally reported, whereas other

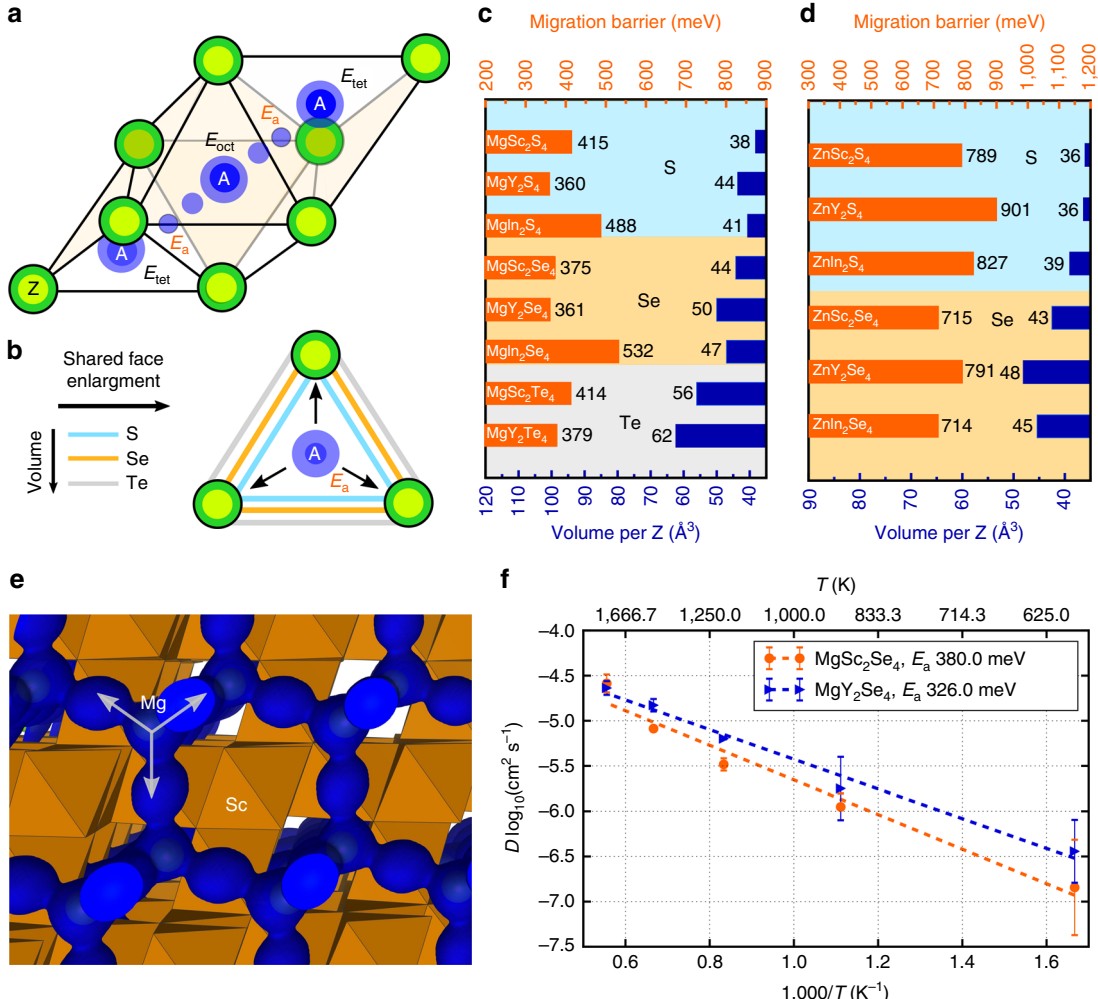

**Fig. 1** First-principles Mg and Zn migration barriers in sulfides, selenides, and tellurides $AX_2Z_4$ spinels (with A = Mg or Zn). **a** *tet–oct–tet* migration path in the $AX_2Z_4$ framework, with energy of the *tet*, *oct*, and transition sites indicated by $E_{tet}$, $E_{oct}$, $E_a$, respectively. $E_a$ corresponds to the migration energy. **b** Effect of the anion size on the shared (triangular) face between *tet* and *oct* sites. **c** and **d** computed Mg and Zn migration barriers (orange bars in meV) in $AX_2Z_4$ spinel and volume per anion (blue bars), respectively, with X = Sc, Y, and In, and Z = S, Se and Te. **e** Mg probability density in $MgSc_2Se_4$ at 900 K obtained from ab initio molecular dynamic simulations (AIMDs). **f** Mg diffusivities as extrapolated from AIMD in $MgSc_2Se_4$ (orange) and $MgY_2Se_4$ (blue), with dashed lines and error bars indicating Arrhenius fits and SD, respectively

sulfides and selenides, such as $MgY_2S_4$, $MgIn_2Se_4$, $ZnSc_2Se_4$, and $ZnIn_2Se_4$, as well as the tellurides ($MgSc_2Te_4$, $MgY_2Te_4$) have not been synthesized to date. The Mg migration barriers, $E_a$ in Fig. 1c indicate that the fastest Mg conductors are $MgY_2S_4$ (~ 360 meV), $MgY_2Se_4$ (~ 361 meV), and $MgSc_2Se_4$ (~ 375 meV). Migration barriers of 361–375 meV ($MgY_2Se_4$ and $MgSc_2Se_4$) signify remarkably high Mg mobility and are comparable to $Li^+$ in fast Li-conductors such as LISICON-like (~ 200–500 meV) and Garnets (~ 400–500 meV)[1, 39]. The Mg migration barriers of $MgY_2Se_4$ and $MgSc_2Se_4$ (Fig. 1c) are further confirmed by AIMD simulations (~ 380 and 326 meV, respectively, Fig. 1f), attesting that selenide spinel frameworks are excellent three-dimensional Mg conductors as shown by the Mg probability density in $MgSc_2Se_4$ of Fig. 1e.

For Mg (and Zn) chalcogenide spinels, the migration barriers $E_a$ (orange bars in Fig. 1c, d) generally decrease with an increase in the volume per anion (blue bars, S < Se < Te). The variation of the metal ion (X in $MgX_2Z_4$) has a marginal effect on the Mg migration barriers, with In-containing compounds displaying larger migration barriers than Sc and Y-containing structures (Fig. 1c). From the data in Fig. 1, it is clear that low Mg migration barriers are achieved by choosing large anions (which guarantees

large triangular faces for $Mg^{2+}$ cations to diffuse through, Fig. 1b), while selecting an appropriate metal (i.e., Sc, Y or In) minimizes the relative energy difference of *oct* and *tet* sites (Supplementary Figs. 1, 2, and 3).

Zn migration barriers are consistently higher in the sulfide and selenide spinels (> 700 meV, Fig. 1d) as compared with their Mg analogs, in agreement with trends in oxide-spinels and the aforementioned design criterion on selecting sites with less favorable coordination[25, 21]. As the preferred anion coordination of $Zn^{2+}$ is 4, the displacement of Zn from *tet* sites (in the *tet–oct–tet* pathway) requires a large energy as quantified by the high migration energy (> 700 meV) of Fig. 1d and Supplementary Fig. 4. These results indicate that $ZnX_2Z_4$ spinels may not be good Zn conductors. For this reason, we focus only on $MgX_2Z_4$ materials, specifically the experimentally reported $MgSc_2Se_4$ and $MgY_2Se_4$, which are predicted to have high Mg mobility based on our NEB and AIMD simulations.

**Synthesis and characterization of magnesium selenides.** $MgSc_2Se_4$ and $MgY_2Se_4$ were initially synthesized by Guittard et al.[34] via a two-step process, starting with the synthesis of the binary selenides (e.g., MgSe and $Sc_2Se_3$), which were

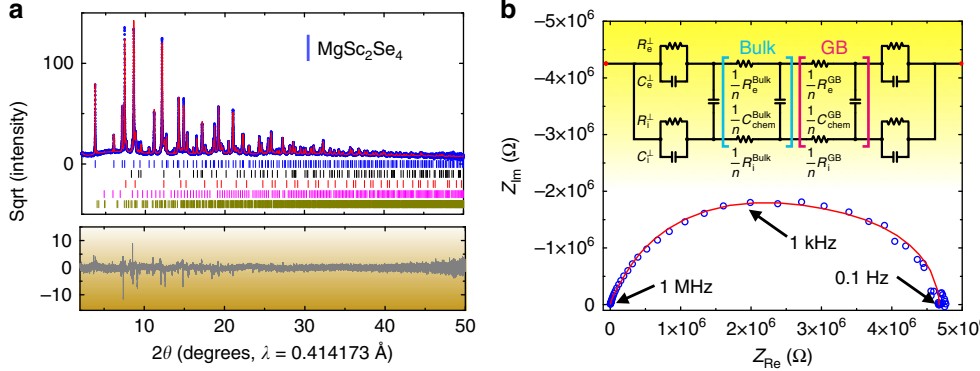

**Fig. 2** X-ray diffractions and electrochemical impedance characterizations of $MgSc_2Se_4$. **a** Rietveld refinement of the synchrotron XRD pattern for $MgSc_2Se_4$. The square root of the intensity is plotted on the y-axis. The observed and calculated curves are shown in blue and red in the top panel, and the difference curve is shown in dark gray in the bottom panel. Reflections corresponding to $MgSc_2Se_4$ (blue), Mg (black), MgSe (red), $Sc_2O_3$ (magenta), and $Sc_2Se_3$ (dark yellow) are shown with tick marks of the respective colors. **b** Impedance spectrum of the $Ta/MgSc_2Se_4/Ta$ cell, and the circuit utilized in the fitting of the impedance data (Supplementary Fig. 10). The observation is shown in blue circles and the fit is displayed in red. The equivalent circuit utilizes two Jamnik–Maier elements[41], which are tentatively attributed to contributions from bulk and grain boundary, respectively. Supplementary Fig. 12 shows the impedance behavior at low $Z_{Re}$

subsequently reacted at 1,200 °C to form $MgSc_2Se_4$ and $MgY_2Se_4$. Nevertheless, neither diffraction patterns nor structural parameters were reported in the original work, whereas ionic transport properties were not characterized[34].

The analysis of the ternary Mg-Sc-Se and Mg-Y-Se phase diagrams, computed with first-principles calculations (Supplementary Fig. 5), reveals that a two-step synthesis may not be necessary. Indeed, the computed formation enthalpies of $MgSc_2Se_4$ and $MgY_2Se_4$ from the pure elements are remarkably negative (~ 1,000 kJ mol⁻¹, Supplementary Table 1 and Supplementary Note 2). In contrast, the formation enthalpy of the spinels from the binaries, whereas negative, is much lower, indicating that this reaction may be slow. After ball milling the elements (Mg, Se, and Sc/Y) the transition metal binary phases ($Sc_2Se_3$ and $Y_2Se_3$) are observed in XRD, but no crystalline MgSe was detected (Supplementary Fig. 6). The $MgSc_2Se_4$ and $MgY_2Se_4$ spinels were obtained after heating the ball milled mixture at 1,000 °C under a flow of Ar (Supplementary Fig. 7).

Although high-purity $MgSc_2Se_4$ can be obtained, the as-prepared $MgY_2Se_4$ samples always contain undesired $Y_2O_2Se$, which complicates the assessment of the intrinsic Mg mobility in $MgY_2Se_4$. In addition, we observed significant inversion while synthesizing spinel-$MgIn_2S_4$ and have investigated in detail the impact of inversion on Mg mobility in a recent work[40]. Therefore, $MgSc_2Se_4$ was chosen for further detailed characterization using synchrotron XRD. The high resolution ($d_{min}$ ~ 0.6 Å) and high counting statistics (signal-to-noise ratio) of the synchrotron XRD data (Fig. 2a) allowed us to determine accurately the structure of $MgSc_2Se_4$, as well as to identify the presence of impurity phases not detectable with laboratory XRD measurements. Through Rietveld refinement of the diffraction data, we conclude that the as-prepared $MgSc_2Se_4$ is uniquely identified as a $MgAl_2O_4$–type spinel (Fd-3m), where the 8b site (tet) is fully occupied by Mg and the 16c site (octahedral) is fully occupied by Sc. Details of the refinement together with the refined structural parameters for $MgSc_2Se_4$ are provided in Supplementary Tables 2–4. The diffraction data of the initially prepared sample also identified a few impurity phases, such as $Sc_2Se_3$ (3.7 wt%), MgSe (7.4 wt%), Mg (3.6 wt%), and $Sc_2O_3$ (1.5 wt%), which suggest that the reaction is incomplete and that a small amount of metal oxide impurities (e.g., MgO and/or $Sc_2O_3$) may be present in the metal precursors. Thus, a subsequent re-sintering at 1,000 °C for ~ 100 h was performed to complete the reaction (the XRD data are

presented in Supplementary Fig. 8), after which the collected powder sample was employed for the assessment of Mg mobility.

**Magnesium mobility.** To obtain direct local Mg ion dynamics and information of the Mg mobility within the $MgSc_2Se_4$ structure, we performed NMR relaxometry and studied the lineshape narrowing via $^{25}Mg$ SS-NMR spectroscopy. A single sharp $^{25}Mg$ NMR lineshape at 53.3 p.p.m. with a quadrupolar coupling constant of only ~ 50 kHz and full width at half maximum of ~ 80 Hz is observed for $MgSc_2Se_4$ at 348 K, shown in Fig. 3a. No other resonances indicating the presence of other Mg environments were detected within ± 20,000 p.p.m. range, consistent with the single crystallographic tetrahedral Mg site in the scandium selenide spinel lattice. A complex and slight NMR line width narrowing (Fig. 3 and Supplementary Fig. 15) has been recorded via variable temperature measurements between 240 K and 470 K. At least two components are needed to de-convolute the NMR peak, one sharp and one broad (the fitting is explained in Supplementary Fig. 16), with the latter component diminishing at higher temperatures. Minor lattice defects or Gaussian broadening of the signal due to the so called "rigid lattice"—where the motional averaging of the signal is inhibited at low temperatures—are likely the causes of the broad band observed[42, 43], whereas the reduction of the band width is related to motional effects. As temperature is varied, no changes in chemical shift are observed, consistent with the diamagnetic nature of the sample, expected from $Sc^{3+}$ with a $d^0$ configuration. Spin lattice relaxation (SLR) times have been collected at various temperatures between 250 and 470 K, to probe $Mg^{2+}$ dynamics. Short SLRs (of 0.07–0.3 s within the measurement temperature range) have been accumulated for the diamagnetic $MgSc_2Se_4$, indicative of significant ionic mobility. For comparison, cubic-MgO or spinel-$MgAl_2O_4$[44] with negligible Mg mobility show sharp NMR lineshapes, but with very slow SLRs, in the order of tens to hundreds of seconds and drastically different relaxometry behavior[45]. Several reports can be found in the literature demonstrating the use of NMR relaxometry analysis to effectively probe the local correlated motion of nuclei under observation, within solid lattices[46–51]. Similarly, NEB simulations provide a description of short range Mg diffusion.[19, 21] By directly applying the analysis described by Kuhn et al.[42, 43] on $MgSc_2Se_4$ $^{25}Mg$ SLR data (further details in the Supplementary Note 4), a maximum mean local Mg jump rate of $1.15 \times 10^8$ Hz at 450 K (assuming that the local maximum is

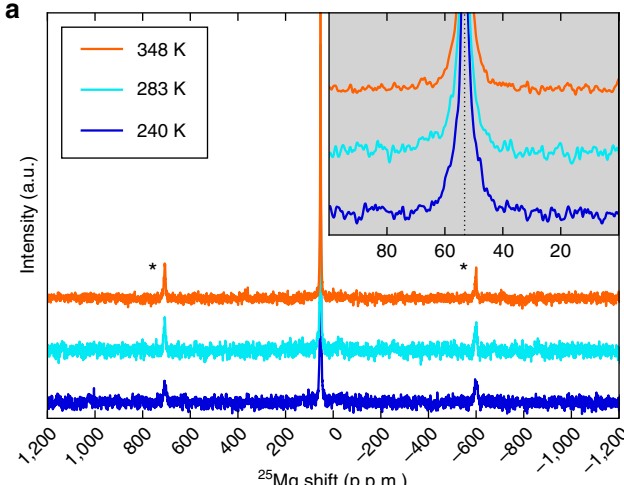

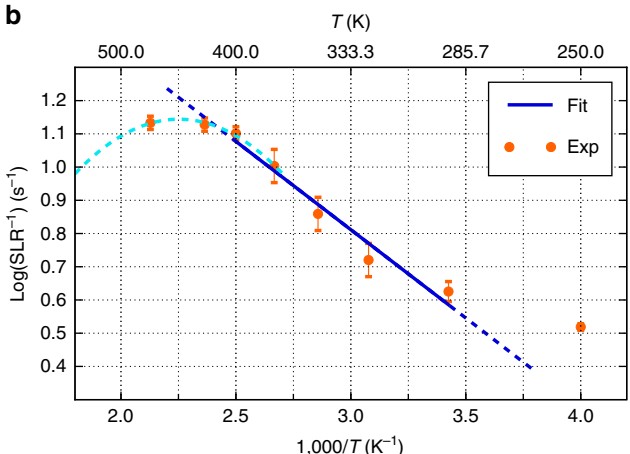

**Fig. 3** Characterization of Mg transport in MgSc$_2$Se$_4$ via $^{25}$Mg solid-state NMR. **a** Stack plot of $^{25}$Mg magic angle spinning (MAS) variable temperature NMR of MgSc$_2$Se$_4$ collected at 11.7 T with a spinning speed of 20 kHz. *Spinning sidebands. **b** $^{25}$Mg static variable temperature spin lattice relaxation data collected at 7.02 T plotted as function of temperature and Arrhenius fit (blue line). Dashed dark blue line illustrates the deviation of experimental data from the fit. The light blue dashed parabolic curve mimics the expected inverse SLR maxima vs. recorded data. The inset of panel **a** shows the enlargement of the $^{25}$Mg MAS NMR signal at ~ 53 p.p.m.

probes local Mg jumps and microscopic diffusion, the assessment of long range motion (or macroscopic diffusion) typically requires static or pulsed field gradient NMR techniques[52], and is not attempted in this study due to major challenges related to $^{25}$Mg nuclei. However, future development of $^{25}$Mg field gradient NMR methods can be envisioned provided suitable materials and isotopic enrichment. The relaxation data below 290 K are omitted since non-diffusive background effects, such as lattice vibrations, predominantly govern the relaxation behavior at low temperatures. The extrapolated room temperature (298 K) ionic conductivity based on the measured activation barrier (370 meV) and Mg jump rate (~ 10$^8$ Hz) at 450 K is ~ 0.01 mS cm$^{-1}$, and in agreement with the conductivity extrapolated by AIMD simulations of ~ 0.089 mS cm$^{-1}$.

Impedance spectroscopy was used to confirm the high Mg mobility observed in $^{25}$Mg magic angle spinning (MAS) NMR measurements (Fig. 2b) utilizing a cold-pressed MgSc$_2$Se$_4$ pellet sandwiched between two identical ion-blocking electrodes. Through a comparison of impedance responses with the use of different blocking electrodes (e.g., Au, Ag, C, In, Pt, Ta, and W), the Ta/MgSc$_2$Se$_4$/Ta configuration was selected. All other blocking electrodes either show reactivity toward the pellet (Ag, C, In, and W; Ag data in Supplementary Information) or require preliminary processing (such as deposition of Au and Pt) in which air exposure is difficult to eliminate. Air exposure causes a dramatic increase in impedance (data of Pt blocking electrode in Supplementary Fig. 9). The impedance spectrum of a Ta/MgSc$_2$Se$_4$/Ta cell, with DC voltage amplitude of 10 mV, as shown in Fig. 2b, is characteristic for mixed ionic-electronic conduction (frequency range, 1 MHz to 0.1 Hz). The interpretation of impedance spectra of mixed conductors has been elaborated in the literature and is conventionally modeled by the generalized circuit of Jamnik and Maier[41, 53]. As a preliminary estimate, the $Z_{re}$ axis intercept at high frequency (or left $x$ axis intercept, ~ 1 MHz) can be approximated as the ionic and electronic resistance contributions in parallel, whereas the intercept at very low frequency (or right $x$ axis intercept, ~ 0.1 Hz) is predominantly the electronic resistance contribution. The validity of this approximation is further verified by analyzing the response of simulated impedance spectra at different values of ionic resistance for the same electronic resistance (Supplementary Fig. 13). A comparison of the simulated spectra with the data measured suggests that the electronic and ionic resistance of MgSc$_2$Se$_4$ are approximatively on MΩ and kΩ scale, respectively. The values of ionic and electronic conductivities were extracted through an equivalent circuit to model the impedance data. When considering only a bulk contribution to the ionic conductivity (i.e., employing only one Jamnik–Maier circuit element), the fit deviates significantly from the observed data (Supplementary Fig. 11). The fit of the impedance data improves significantly when coupling two Jamnik–Maier circuit elements in series (Fig. 2b and Supplementary Fig. 10), one for the bulk response and the other for grain boundary contributions (details of the fitting procedure and parameters in Supplementary Note 3). It is noteworthy that the current data set does not allow us to unambiguously distinguish between the bulk and grain boundary contributions, the sum of the bulk and grain boundary resistance was therefore taken to calculate the ionic conductivity. Although high Mg mobility is observed (ionic conductivity of ~ 0.1 mS cm$^{-1}$ at 298 K), the electronic conductivity of MgSc$_2$Se$_4$ is ~ 0.04 % of the ionic conductivity, which is substantially larger than in other state-of-the-art alkali solid-state electrolytes ($\sigma_e/\sigma_i$ ~ 10$^{-4}$–10$^{-6}$ %)[1, 54, 55]. Although the impedance data at elevated temperature becomes increasingly scattered due to pronounced reactivity of MgSc$_2$Se$_4$ against Ta blocking electrodes (Supplementary Fig. 14), the extrapolated Mg migration barrier, from variable-temperature

reached at 450 K, only an extrapolation within the margin of the error bars is shown with a dashed light blue curve) can be determined from the recorded diffusion-induced SLR time maxima in Fig. 3b. Using a Mg–Mg jump distance of ~ 4.815 Å (to extract the hop frequency in the pre-exponential term of the Arrhenius equation controlling Mg diffusion) as determined from the synchrotron XRD data (Supplementary Tables 2, 3 and 4), the local Mg jump rate corresponds to a maximum self-diffusion coefficient of ~ 4.53 × 10$^{-8}$ cm$^{-2}$ s$^{-1}$ at 450 K. A linear Arrhenius fit to the relaxation data in the ~ 320–400 K temperature range (solid blue line in Fig. 3b)[42, 43], gives a Mg migration barrier of 370 ± 90 meV, in excellent agreement with the computed data of MgSc$_2$Se$_4$ obtained with NEB (~ 375 meV, Fig. 1c) or extrapolated from Arrhenius' fit (~ 380 meV) of the Mg diffusivities from AIMD simulations (Fig. 1e). This is also in agreement with values obtained from motional linewidth narrowing analysis described in Supplementary Note 4. However, the activation energies determined experimentally correlates solely to the local/short range motion of Mg ions within the lattice and is generally smaller than long range motion. Although NMR relaxometry

impedance measurement, is ~ 200 ± 40 meV. The electrochemical stability window of $MgSc_2Se_4$ is also assessed both experimentally (Supplementary Fig. 20 and Supplementary Note 8) and theoretically (Supplementary Fig. 18).

## Discussion

In summary, we report the first demonstration of fast Mg-ion conduction in close-packed frameworks, specifically in spinel-$MgSc_2Se_4$. The spinel structures were chosen based on the design criteria that Mg mobility is highest in structures with Mg in unfavorable coordination and with high volume per anion[20, 21, 25, 27, 31]. Mg NMR relaxometry and impedance spectroscopy confirm the fast $Mg^{2+}$ motion with a low migration barrier (~ 370 ± 90 meV). First-principles calculations and ab initio molecular dynamics indicate that several other spinels in this family are likely to also have high Mg mobility, including $MgY_2S_4$ (~ 360 meV) and $MgY_2Se_4$ (~ 361–326 meV). These migration barriers compare well against those of $Li^+$ in fast Li-ion conductors, such as Garnets (~ 400–500 meV)[39], and are substantially lower than other Mg-conductors reported to date[39]. As the impedance spectroscopy indicates mixed conduction behavior with appreciable electronic conductivity, strategies to suppress the electronic conductivity should be sought for the material to become a practical solid-state Mg electrolyte.

The origins of the electronic conductivity observed can be related to: (i) the existence of intrinsic defects, e.g., Mg, Sc, or Se vacancies, or (ii) the presence of undesired electron conducting secondary phases in the as-prepared $MgSc_2Se_4$ sample. The $^{25}Mg$ NMR results do not indicate any significant Mg containing compounds (including $MgSc_2Se_4$) displaying appreciable electronic conduction. Furthermore, the band gap of $MgSc_2Se_4$ is estimated to be ~ 2.15 eV by HSE06[56, 57] calculations (Supplementary Fig. 17 and Supplementary Notes 5 and 6), suggesting that the electronic conductivity of a defect-free $MgSc_2Se_4$ compound should be extremely low. Together, these observations suggest that the electronic conductivity, as observed in $MgSc_2Se_4$, is either caused by the presence of intrinsic defects or by secondary non-Mg containing phases. Understanding the defect chemistry in selenide spinels is therefore crucial to minimize the electronic conductivity. An alternative strategy to circumvent electronic conduction is to engineer the surface of $MgSc_2Se_4$ to be electron-insulating but ion-conductive, which can be achieved either through ex situ coating of a thin-layer of a different material or via in situ formation of a thin interface between electrodes and the $MgSc_2Se_4$ solid-state electrolyte (Supplementary Fig. 19).

Practical coating layers need to show sufficient Mg mobility to ensure good performance of an all-solid-state battery. Indeed, we considered Mg diffusion across prominent electrolyte decomposition products against Mg metal, namely the binary MgO, MgS, and MgSe. We found high diffusion barriers (> 800 meV, Supplementary Fig. 19, and Supplementary Note 7) in MgO and MgS, while MgSe exhibits lower value (~ 695 meV). Thus, potential Mg solid electrolytes, especially those that are composed of oxides and sulfides, will need to ensure the formation of interfacial products with better Mg mobility when used against Mg metal, compared to the binary Mg chalcogenides. We further note that the challenge of interfaces to almost all solid-state battery technologies, including alkali-based systems[58, 59].

Besides identifying the first spinel, $MgSc_2Se_4$ (and $MgY_2Se_4$), with high room temperature Mg ionic conductivity our work also validates the previously identified design rules for fast multivalent-ion solid conductors[20, 21], and present an encouraging step towards finding more solids with fast $Mg^{2+}$ mobility that can function as electrode or electrolyte materials.

## Methods

**Ion diffusion**. Mg and Zn migration barriers are assessed using the NEB[35] method in combination with DFT[60], whereas approximating the exchange correlation with the Perdew–Burke–Ernzerhof generalized gradient approximation[61], as implemented in the Vienna Ab initio Simulation Package[62, 63]. We used a $2 \times 2 \times 2$ supercell of the primitive spinel cell for the NEB calculations, corresponding to 64 anion atoms. The total energy was sampled on a well-converged $2 \times 2 \times 2$ $k$-point grid together with projector-augmented wave theory[64] and a 520 eV plane-wave cutoff, and converged within $1 \times 10^{-5}$ eV per supercell. Mg (and Zn) migration in the chalcogenides is assessed in the low-vacancy limit –one Mg (Zn) vacancy per supercell– and S, Se, or Te excess electrons are balanced with a uniform background charge. A previous report has verified that the background correction to compensate for a Li vacancy (in $Li_2S$), does not alter significantly the electronic charge density of the host material as well as the magnitude of the activation barriers related to ion diffusion as compared with chemical doping[20]. The endpoint structures were fully relaxed until the forces on atom converged within $1 \times 10^{-2}$ eV $Å^{-1}$, whereas the NEB forces were converged within 0.05 eV $Å^{-1}$. Introducing a minimum distance of at least 10 Å between the Mg (Zn) ions minimizes fictitious interactions across periodic boundaries. Nine distinct images are used between the endpoints to evaluate the ion migration trajectory.

The accuracy of the NEB simulations was contrasted by AIMD simulations within the Born–Oppenheimer approximation, which give access to Mg diffusivities in $MgSc_2Se_4$ and $MgY_2Se_4$ as a function of temperature. The atom trajectories are propagated with the Verlet scheme within the canonical NVT ensemble, relying on the Nosé–Hoover thermostat with a period of 120 and 2 ps time step. Mg self-diffusivities are obtained by using the Einstein relation, fitting mean-squared displacements against time. Subsequently the Mg migration energies ($E_a$) are extrapolated by an Arrhenius fit of the diffusivity data vs. temperature (see Fig. 1f). In AIMD simulations, both total energy and forces are sampled on a single $k$-point.

**Synthesis and synchrotron diffraction of $MgSc_2Se_4$ and $MgY_2Se_4$**. Elemental forms of Mg (Sigma Aldrich, ≥ 99%), Sc (Sigma Aldrich, 99.9%), or Y (Goodfellow, 99.9%), and Se (Sigma Aldrich 99.99% trace metal basis) are first weighted with the stoichiometric ratio. Approximately 3.0 g of the powder mixture was placed into a tungsten carbide ball mill jar, which was ball milled (SpexSamplePrep 8000 M) for 30 min. The resulting powder was then pressed into pellets of 6.0 mm in diameter under a pressure of 1.4 metric tons for 1–2 min. Typically, 2–3 pellets, each weighted ~ 0.2 g, were wrapped into a platinum foil (Sigma Aldrich, 99.99% trace metal basis), which was subsequently secured into a stainless-steel tube (Swagelok, 3/8-inch diameter). The tube was later closed with stainless steel caps, to avoid air exposure during the transfer of the tube for synthesis. All the aforementioned steps were performed in an Ar glove box. The reaction was carried out in a Thermo Scientific Minimite furnace under a continuous flow of Ar gas. To further reduce the level of oxygen and moisture in the Ar gas, an oxygen/moisture trap was attached between the Ar gas cylinder and the quartz tube where the stainless-steel tube was placed. After a quick purge of the tube (~ 20 min), the temperature was quickly ramped to 1,000 °C in 1 h. The temperature was held at 1,000 °C for 12 h, before the furnace was naturally cooled down to room temperature. The powder was collected in Ar glove box by cutting the stainless-steel tube.

To verify the phase purity of the as-prepared $MgSc_2Se_4$ and $MgY_2Se_4$ samples, XRD was performed on a Rigaku Miniflex 600 diffractometer with Cu $K_\alpha$ radiation. For structural determinations, the diffraction data was collected at the beamline 11BM at the Advanced Photon Source of Argonne National Laboratory with a constant wavelength of ~0.41 Å. Owing to the air sensitivity of the selenide sample, the as-prepared $MgSc_2Se_4$ powder was packed into a 0.5-mm diameter special glass capillary in an argon glove box. The glass capillary was further secured into a 0.8-mm diameter Kapton tube. The Rietveld refinements were performed using the TOPAS 4.2 software package (Bruker).

**Solid-state variable temperature nuclear magnetic resonance**. $^{25}Mg$ MAS NMR experiments were performed at 11.7 Tesla (500 MHz), while static variable temperature $^{25}Mg$ relaxometry experiments were performed at 7.02 Tesla (300 Mhz) on a Bruker Avance III spectrometer operating at a Larmor frequency of 30.64 MHz (for 11.7 Tesla) and 18.37 MHz (for 7.02 Tesla), respectively. All samples were packed under a continuous flow of Ar. A calibrated $\pi/2$ (actual $\pi/6$) pulse width of 3 µs was used. The MAS spectra were acquired at a spinning speed of 20 kHz using 3.2 mm rotors (7 mm rotors for static measurements) with a rotor synchronized spin-echo experiment ($90°–\tau–180°–\tau$) where $\tau= \nu_r^{-1}$. For low field variable temperature measurements, single pulse experiments with recycle delays of 0.1 s, 0.2 s, 0.5 s, 1 s, 2 s, and 5 s. Calibrated $\pi/2$ (actual $\pi/6$) pulse widths of 2.92 µs were used. As saturation recovery experiments failed to be effectively applicable for $^{25}Mg$ relaxometry, logarithmic fits with temperature dependent statistical errors (error bars plotted in Fig. 3b, representative fits shown in Supplementary Fig. 14) were obtained to determine best SLR time estimates at 50% maximum intensity vs. 10 s recycle delay value. Similar trends with slightly larger SLRs (0.3 to 0.9 s) and slightly lower activation energies within the reported measurement errors were reproduced at 63% maximum intensity vs. 60 s recycle delay value. All chemical shifts were referenced to an aqueous solution of 5 M $MgCl_2$ at 0 p.p.m.

**Impedance spectroscopy**. Impedance measurements of $MgSc_2Se_4$ were performed with a Solartron MTS system using Swagelok cells. The impedance data were collected from 1 MHz to 0.1 Hz with DC voltage amplitude of 10 mV. To prepare the sample for impedance measurements, 80–100 mg of $MgSc_2Se_4$ powder was first pressed into a disk-shaped pellet with the use of a 6.0 mm die under a pressure of 1–1.2 metric tons. The resulting pellet, typically with the thickness of 0.5–1.0 mm, was assembled into a spring-loaded Swagelok cell, using stainless steel rods as current collectors covered with tantalum foils (~0.05 mm thick) (Sigma Aldrich, ≥ 99.9%). The preparation of the sample and the Swagelok cell was entirely completed in an Argon glove box.

**Data availability**. The data to support the findings of this study are available from the corresponding authors upon request.

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

## Acknowledgements

This work was fully supported as part of the Joint Center for Energy Storage Research (JCESR), an Energy Innovation Hub funded by the U.S. Department of Energy, Office of Science, and Basic Energy Sciences. This study was supported by subcontract 3F-31144. We also thank the National Energy Research Scientific Computing Center (NERSC) for providing computing resources. Use of the Advanced Photon Source at Argonne National Laboratory was supported by the U.S. Department of Energy, Office of Science, Office of Basic Energy Sciences, under Contract No. DE-AC02-06CH11357. We acknowledge Dr. Niya Sa and Dr. Sang-Don Han at Argonne National Laboratory for their experimental support regarding Mg electrochemistry measurements. We are grateful to Dr. Venkat Srinivasan at Lawrence Berkeley National Laboratory for stimulating discussions on the interpretation of impedance data and to Dr. Sylvio Indris at Karlsruhe Institute of Technology for discussions on NMR relaxometry.

## Author contributions

P.C. and S.-H.B. designed the study and performed the first-principles calculations. S.-H. B. designed the synthesis including XRD and impedance characterization, and performed the electrochemistry characterizations together with T.S. and Y.T. B.K. performed the [25]Mg NMR characterization and analysis. P.C., S.-H.B., G.S.G. and W.D.R. jointly performed all data analysis and wrote the manuscript together with G.C. Y.W., W.D.R. and J.C. provided insights for the execution of the first-principles calculations and experiments. All authors approved the final version of the manuscript.

## Additional information

**Competing interests:** The authors declare no competing financial interests**Supplementary Information** accompanies this paper at doi:10.1038/s41467-017-01772-1.

