## [Peer Review File · Nature Communications]

Reviewers' comments:

Reviewer #1 (Remarks to the Author):

This paper reports the discovery of a novel Mg-ion solid conductor with exceptionally high Mg diffusivity at room temperature. This is a very important finding in Mg battery research. The paper is very nicely written and combines computational, solid-state NMR, and impedance measurement/modeling to address a challenge that has previously been considered impossible in the materials science community. It is a transformative contribution to the rapidly evolving field.

Before it can be published, the authors must address the following questions/comments.

1. Fig. S20 is a LSV curve instead of a CV curve. Please make sure the CV plot is correctly used.
2. Interlayer expansion approach for layered host materials have been previously demonstrated to reduce Mg²⁺ diffusion barrier and increase its diffusivity. A relevant reference is recommended to be cited: Liang, Y. et al. Nano Lett. 2015, 15, 2194-2202.

Reviewer #2 (Remarks to the Author):

1. "Identifying the one (or several) material(s) which show(s) high Mg mobility from the ~ 10,000 reported Mg containing compounds should be considered as a major leap by itself." I fully agreed with this point, while the authors did not show how they identify this. It will be great that the authors show the calculated/experimental data to exhibit how they screen the one or several materials from 10,000 compounds.
2. The authors mentioned that the background charge is used to balance the charge of vacancy during calculation of ion diffusion. Will this affect the diffusion barrier? The authors should explain more on this. Meanwhile will the concentration of the vacancy affect the diffusion?
3. It is well-known that the GGA underestimates the barriers compared with the experimental results. As mentioned by the authors their DFT results agree remarkable well with the experiments. Does the diffusion barrier affect the functional used?

Reviewer #3 (Remarks to the Author):

The paper entitled „High Mg²⁺ Cation Mobility in Solids“ contains a combined theoretical and experimental study for Mg ion diffusion in Mg conductors, especially in MgSc₂Se₄. The manuscript is not worth for publication in Nature Communications as it contains huge drawbacks. The contents of the manuscript are less suited for the scopes of this journal. The applied methods (at least the theoretical approaches) are too poor to get published in any high profile journal. In the following, the drawbacks of the paper are listed to help the authors:

1. The authors mentioned the topic as “High Mg²⁺ Cation Mobility in Solids”, however the most part of the study contains results on MgSc₂Se₄. It seems that they want to “Make a mountain out of a molehill”.
2. The main conclusion is based on the theoretical investigations of activation energy calculated by DFT/NEB methods. By comparing the activation energy values for some Mg (Zn) chalcogenides, they have limited their further studies on MgSc₂Se₄ material only. The policy is acceptable probably,

however the applied methods are not scientifically of high level. In the later part, the authors have correctly p

edicted that pure DFT approaches fail to reproduce the band gaps of Mg chalcogenides, rather they have utilized the HSE06 approach for this purpose. The study should be uniform: HSE06 approach should be used for all the parts of study from the very beginning. Based on the available literature data on ion diffusion in solids, it is known that pure DFT approaches underestimate the experimental (microscopic diffusion) activation barriers whereas the HF/DFT hybrid approaches give close agreement to the experimental values. Therefore, 'very close agreement' of the DFT activation barriers with the experimental values will create huge doubt in the readers' minds. The authors should have also investigated the frequency calculations to verify the saddle points and minima.

3. In the structural characterization (S2), the authors have predicted enthalpy of formation at 0K, it is rather useless. They should have done high temp Freq based calculations to draw the conclusion at high temp or at least at room temperature. By this way, the conclusions could be different.

4. In the SLR investigation, it is shown that the magnetism plays a key role, as can be seen that the diamagnetic MgSc₂Se₄ has fast ionic conductivity. There is no theoretical investigation of magnetism of the considered materials.

5. The considered models for the theoretical investigation are not 'ideal ones'. The authors have models the balance of excess electrons by a uniform background charge. Rather the electrons should be treated with open shell spin polarized methods and of course, the magnetism for all the cases should be checked by energy/enthalpy calculations.

6. The size of the supercell is not clear. In experiment, the vacancy concentration is very low. In order to model this, they should have used a very large supercell.

In general the quality of this paper is too poor. I recommend an immediate rejection of this paper. No further review required.

Subject: Decision on manuscript NCOMMS-17-08731-T

Dear Editor,

Thank you for sending us the referee comments on our manuscript. We are grateful to all the referees for bringing up important points that can improve the overall quality and readability of this novel and original work. Based on the suggestions of the reviewers, changes have been introduced to further strengthen our manuscript. Please find below a detailed response to all the comments and suggestions raised by the reviewers. In general, we appreciate the very positive comments of both reviewers 1 and 2. Many of the comments and criticisms reviewer 3 brings up relate to well established methods and procedures in the field of computational materials science, and energy storage in particular. Hence, we argue that reviewer 3 is not qualified, and shows remarkable bias against the work, as evidenced by the language used. We hope that the revised manuscript will convince the referee(s) and the editor of the timely importance of this work for the development of materials with high Mg mobility, resulting in a swift publication.

Reviewer 1:

This paper reports the discovery of a novel Mg-ion solid conductor with exceptionally high Mg diffusivity at room temperature. This is a very important finding in Mg battery research. The paper is very nicely written and combines computational, solid-state NMR, and impedance measurement/modeling to address a challenge that has previously been considered impossible in the materials science community. It is a transformative contribution to the rapidly evolving field.

We appreciate Reviewer 1, who identifies the importance of this work– *“the discovery of a novel Mg-ion solid conductor with exceptionally high Mg diffusivity at room temperature.”*

Before it can be published, the authors must address the following questions/comments.

1. *Fig. S20 is a LSV curve instead of a CV curve. Please make sure the CV plot is correctly used.*

The revised supplementary information now uses the appropriate terminology, i.e. linear sweep voltammetry, when discussing **Figure S20a**.

2. *Interlayer expansion approach for layered host materials have been previously demonstrated to reduce Mg²⁺ diffusion barrier and increase its diffusivity. A relevant reference is recommended to be cited: Liang, Y. et al. Nano Lett. 2015, 15, 2194-2202.*

The revised manuscript (see the Introduction section) includes a sentence and the citation suggested by the reviewer, explaining that ion diffusion can be enhanced by expanding the interlayer distance in layered materials.

Reviewer 2:

1. *“Identifying the one (or several) material(s) which show(s) high Mg mobility from the ~ 10,000 reported Mg containing compounds should be considered as a major leap by itself.” I fully agreed with this point, while the authors did not show how they identify this. It will be great that the authors show the calculated/experimental data to exhibit how they screen the one or several materials from 10,000 compounds.*

We are grateful that Reviewer 2 recognizes that we have found a fast Mg conductor. The original manuscript summarizes the 3 main criteria followed to short-list the Mg conductors studied in this paper, which are: i) find materials where Mg resides

in its less preferred anion coordination environment (i.e. coordination of 4), *ii*) find materials with large volume per anion (typically introduced by anions with large ionic radii, e.g. S, Se and Te), and *iii*) find materials without redox-active metals. The revised Supplementary Information contains a new section, Section S9 Mg conductors screening, which explains in detail the procedure used to locate fast Mg conductors according to the 3 criteria.

2. *The authors mentioned that the background charge is used to balance the charge of vacancy during calculation of ion diffusion. Will this affect the diffusion barrier? The authors should explain more on this. Meanwhile will the concentration of the vacancy affect the diffusion?*

Use of the background charge: Using a uniform background charge to compensate for charged defects is a well-established procedure in *ab initio* defect calculations. The uniform background charge is routinely used in theoretical studies assessing the formation of point defects in semiconductor materials, as indicated by Freysoldt *et al.*, “*First-principles calculations for point defects in solids*”, *Rev. Mod. Phys.*, **86**, 253 (2014), doi: 10.1103/RevModPhys.86.253 and Freysoldt *et al.*, *Phys. Rev. Lett.*, **102**, 016402 (2009), doi 10.1103/PhysRevLett.102.016402]. A recent research article published by some of the authors [Y. Wang *et al.*, “*Design principles for solid-state lithium superionic conductors*”, *Nature Materials*, **14**, 1026-1031 (2015), doi: 10.1038/nmat4369], furthermore demonstrates (see text and Figures S13, S14 and S15 in the Supplementary Information of Y. Wang *et al.*) that the charge density of a charge-compensated Li₂S cell and a chemically doped cell (with Mg²⁺) do not vary significantly, thereby confirming that the background charge correction does not alter the energy landscape for ion diffusion. The method section of the revised manuscript now contains an additional sentence discussing this aspect.

Effect of vacancy concentration: In a recent independent study, X. Sun *et al.*, [“*A high capacity thiospinel cathode for Mg batteries*”, *Energy Environ. Sci.*, 2016, **9**, 2273-2277, doi: 10.1039/C6EE00724D] suggested that the activation barriers for Mg migration are concentration independent, reporting values of ~600 meV and ~550 meV in the dilute vacancy and the concentrated limits, respectively. The difference of ~50 meV, falls within the domain of the Nudged Elastic Band accuracy. However, the referee should note that this similarity in activation energies is only true for chalcogenides spinels, and substantial differences are observed in oxide spinels, such as MgMn₂O₄, as documented by Rong *et al.*, “*Materials Design Rules for Multivalent Ion Mobility in Intercalation Structures*”, *Chem. Mater.*, **27** (17), 6016 (2015) doi: 10.1021/acs.chemmater.5b02342. The methods section also includes a sentence explaining this aspect.

3. *It is well-known that the GGA underestimates the barriers compared with the experimental results. As mentioned by the authors their DFT results agree remarkable well with the experiments. Does the diffusion barrier affect the functional used?*

To answer the referee’s comment, we compare in **Table 1** the experimental and the GGA computed activation energies (**E_a**) and ionic conductivities (**σ**) for Li and Na diffusion in several solid electrolytes. **Table 1** clearly shows that the agreement between theory and experiment is not fortuitous, suggesting that GGA activation energies are indeed reliable. Note that all the systems listed in Table 1 consist of closed-shell metals or semi-metals, such as Ge, Sn, La, etc.

Table 1 Experimental and GGA predicted activation energies (**E_a**) and ionic conductivities (**σ**) at 298 K for several Li and Na ionic conductors.

Material	Source	E _a (eV)	σ 298K (mS/cm)	Reference
Li ₉ S ₃ N	Exp.	0.52	8.30E-04	Miara et al. , J. Mater. Chem. A , 3 , 20338-20344 (2015).
Li ₉ S ₃ N	DFT	0.53	2.40E-03	Miara et al. , J. Mater. Chem. A , 3 , 20338-20344 (2015).

Li ₁₀ GeP ₂ S ₁₂	Exp.	0.25	12	Kamaya et al. , Nature Materials , 10 , 682–686 (2011).
Li ₁₀ GeP ₂ S ₁₂	DFT	0.21	13	Ong et al. , Energy Environ. Sci. , 6 , 148-156 (2013).
Li ₁₀ SnP ₂ S ₁₂	Exp.	0.27	4	Bron et al. J. Am. Chem. Soc. 135 , 15694 (2013).
Li ₁₀ SnP ₂ S ₁₂	DFT	0.24	6	Ong et al. , Energy Environ. Sci. , 6 , 148-156 (2013).
Na ₁₀ SnP ₂ S ₁₂	Exp.	0.356	0.4	Richards et al. Nature Communications , 7 , 11009 (2016).
Na ₁₀ SnP ₂ S ₁₂	DFT	0.317	0.9	Richards et al. Nature Communications , 7 , 11009 (2016).
Li _{7.08} La _{2.96} Rb _{0.04} Zr ₂ O ₁₂	Exp.	0.29	0.8	Lee et al. , MRS Meeting Abstract , J13.02 (2012).
Li _{7.08} La _{2.96} Rb _{0.04} Zr ₂ O ₁₂	DFT	0.21	2.74	Miara et al. , Chem. Mater. , 25 , 3048–3055 (2013).
Li ₁₀ SiP ₂ S ₁₂	Exp.	0.196	2.3	Whiteley et al. , J. Electrochem. Soc. , 161 , A1812 (2014).
Li _{9.54} Si _{1.7} P _{1.44} S _{11.7} Cl _{0.3}	Exp.	–	25	Kato et al. , Nature Energy , 1 , 16030 (2016).
Li ₁₁ SiP ₂ S ₁₂	Exp.	0.19		Kuhn et al. , Phys Chem Chem Phys , 16 , 14669 (2014).
Li ₁₀ SiP ₂ S ₁₂	DFT	0.2	23	Ong et al. , Energy Environ. Sci. , 6 , 148-156 (2013).

Figure 1 shows a correlation plot of experimental vs. PBE-GGA computed activation energies, demonstrating further that the level of approximation adopted for our calculations captures the appropriate physics of ion diffusion in these solid electrolytes.

Reviewer 3:

The paper entitled “High Mg²⁺ Cation Mobility in Solids“ contains a combined theoretical and experimental study for Mg ion diffusion in Mg conductors, especially in MgSc₂Se₄. The manuscript is not worth for publication in *Nature Communications* as it contains huge drawbacks. The contents of the manuscript are less suited for the scopes of this journal. The applied methods (at least the theoretical approaches) are too poor to get published in any high-profile journal. In the following, the drawbacks of the paper are listed to help the authors:

We regret that reviewer 3 sees no positive value at all in our work, and thinks of the applied methods as “too poor”. We believe that the reviewer is erroneous on this matter.

1. The authors mentioned the topic as “High Mg²⁺ Cation Mobility in Solids”, however the most part of the study contains results on MgSc₂Se₄. It seems that they want to “Make a mountain out of a molehill”.

Contrary to the referee comment, the manuscript computationally covers multiple spinel chemistries, i.e. AX₂Z₄ structures (with A = Mg or Zn, X= Sc, Y and In, and Z = S, Se and Te). To assess Mg mobility experimentally in this class of materials, we focused on MgSc₂Se₄ and MgY₂Se₄, due to the low Mg migration barriers calculated theoretically (~360 meV). While the manuscript covers in detail the experimental on MgSc₂Se₄ as one example of this class of materials, the manuscript and the supporting information also extensively documents our attempts at synthesizing a phase-pure MgY₂Se₄, which seems to have been overlooked by the referee! The experimental data clearly supports the computational predictions and confirms that high Mg²⁺ mobility is possible. We do not believe that this making a “mountain out of a molehill”

2. The main conclusion is based on the theoretical investigations of activation energy calculated by DFT/NEB methods. By comparing the activation energy values for some Mg (Zn) chalcogenides, they have limited their further studies on MgSc₂Se₄ material only. The policy is acceptable probably, however the applied methods are not scientifically of high level. In the later part, the authors have correctly predicted that pure DFT approaches fail to reproduce the band gaps of Mg chalcogenides, rather they have utilized the HSE06 approach for this purpose. The study should be uniform: HSE06 approach should be used for all the parts of study from the very beginning. Based on the available literature data on ion diffusion in solids, it is known that pure DFT approaches underestimate the experimental (microscopic diffusion) activation barriers whereas the HF/DFT hybrid approaches give close agreement to the experimental values. Therefore, ‘very close agreement’ of the DFT activation barriers with the experimental values will create huge doubt in the readers’ minds. The authors should have also investigated the frequency calculations to verify the saddle points and minima.

We strongly disagree with these reviewer’s statements, all of which are unsupported by even a single reference. There are several flaws in his/her arguments, and we would argue that these arguments are “too poor” to constitute a knowledgeable and unbiased review.

- a) In contrast to the referee’s views, the main conclusion of this paper is not solely based on activation barriers from theoretical predictions. Indeed, we have employed sophisticated variable temperature ²⁵Mg NMR and impedance spectroscopy measurements to verify our theoretical calculations of high Mg²⁺ mobility in MgSc₂Se₄. The referee must also note that the objective of this paper is not to improve existing theoretical methods and/or experimental techniques, but the discovery, for the first time, of high divalent cation mobility in close-packed solids. Hence the comment of the referee, “the applied methods are not scientifically of high level” is unfair and unwarranted, particularly in light of the supportive comments by reviewers 1 and 2. The theoretical methods used in the paper are very commonly used to investigate the phase stability and mobility in ionic conductors. For example, some previous work from my group 1) *Nature Materials*, **14**, 1026-1031 (2015), doi: 10.1038/nmat4369, 2) Ong *et al.*, *Energy Environ. Sci.*, **6**, 148-156 (2013) or 3) Richards *et al.* *Nature Communications*, **7**, 11009 (2016). Most importantly similar techniques have been applied from other groups: i) Chunsheng Wang in *Adv. Energy Mater.*, **6** (8) 1501590, (2016) 10.1002/aenm.201501590, ii) or from the group of Prof. Holzwarth *Phys. Rev. B* **88**, 104103, doi: 10.1103/PhysRevB.88.104103, *Phys. Rev. B* **81**, 184106, doi: 10.1103/PhysRevB.81.184106, or iii) Dr Boris Kozinsky at Bosch USA *Phys. Rev. Lett.* **116**, 055901, doi: 10.1103/PhysRevLett.116.055901, or iv) Dr Wagemaker *Chem. Mater.* 2016, **28**, 7955, doi: 10.1021/acs.chemmater.6b03630.

- b) The argument of reviewer 3, “*it is known that pure DFT approaches underestimate the activation barriers*” is not correct. In **Figure 1** and **Table 1**, we have demonstrated that GGA-based activation barriers can reliably predict activation barriers in several Li- and Na-solid electrolytes, all of which consist of closed-shell metals, in our response to point 3 of reviewer 2’s comments. There is no basis to assume that hybrid functionals would do better in reproducing barriers.
- c) The reviewer suggestion that HSE should be used for the complete study is wrong. Each functional has its own issue, which is why we use different functionals to obtain different properties. It is part of the skill in doing computational materials science to understand the benefits and limitations of each approach in addressing various scientific questions. Hence, until there is a perfect functional, the reviewer’s suggestion that a study must be performed uniformly with a single functional is poor advice, and has no grounds in any facts.

3. In the structural characterization (S2), the authors have predicted enthalpy of formation at 0K, it is rather useless. They should have done high temp Freq based calculations to draw the conclusion at high temp or at least at room temperature. By this way, the conclusions could be different.

The approach of calculating 0K enthalpies to get a sense of phase stability is ubiquitous in first principles materials science, to the point where there are multiple hundreds of papers that use the approach, and some representative publications are listed at the end of this letter.¹⁻¹⁶ So it is definitely not “rather useless” as the reviewer states. In addition, experimental facts in this class of materials support our approach, given that the MgSc₂Se₄ spinel considered in our manuscript has been previously synthesized for superconductor applications, indicating that their formation is favorable thus supporting our DFT findings.

4. In the SLR investigation, it is shown that the magnetism plays a key role, as can be seen that the diamagnetic MgSc₂Se₄ has fast ionic conductivity. There is no theoretical investigation of magnetism of the considered materials.

Since the metals of all the compounds considered (In, Sc, and Y) exhibit closed-shell electronic configurations (d^0 for Sc/Y and d^{10} for In), magnetism is not a concern for evaluating Mg mobility in these compounds. Indeed, no ²⁵Mg-NMR resonances were found within a +/- 20000 ppm range while scanning MgSc₂Se₄ samples, indicating the absence of any paramagnetic Mg environment. In addition, the signal found at 53.3 ppm was not found to shift significantly with temperature, ruling out paramagnetic effects due to the well-known temperature dependence of Fermi-contact shifts or signatures of a Korringa relationship typically found in metals. These experimental findings suggest that the theoretical investigation of magnetism is clearly out of the scope of this work.

5. The considered models for the theoretical investigation are not ‘ideal ones’. The authors have modeled the balance of excess electrons by a uniform background charge. Rather the electrons should be treated with open shell spin polarized methods and of course, the magnetism for all the cases should be checked by energy/enthalpy calculations.

The reviewer seems to grasp at straws here in a his/ her rather unprofessional attempt to discredit our work. Magnetism is not all relevant here as all ions are closed shell ions. We have addressed the issue of using uniform background charges in question No. 2 of referee 2. The background correction has been routinely used for more than 20 years in theoretical studies assessing the formation of point defects in solids and well documented in reviews such as, e.g. C. Freysoldt *et al.*, “*First-principles calculations for point defects in solids*”, *Rev. Mod. Phys.*, **86**, 253 (2014), doi: 10.1103/RevModPhys.86.253. Additionally, a recent research article published by some of the authors [Y. Wang *et al.*, “*Design principles for solid-state lithium superionic conductors*”, *Nature Materials*, **14**, 1026-1031 (2015), doi: 10.1038/nmat4369], clearly demonstrates that

the charge density of a charge-compensated Li_2S cell and a chemically doped cell (with Mg^{2+}) do not vary significantly. Thus, modeling Mg migration in the spinels considered in this work, with a compensating background charge, should lead to an accurate evaluation of the migration barriers. If the reviewer has issues with these well-established method, as he/she seems to have with many other well established methods, I suggest that he/she writes a paper documenting his/her gripes with these methods so that the objections, if they exist at all, can enter the arena of scientific discussion, rather than anonymous review reports.

6. *The size of the supercell is not clear. In experiment, the vacancy concentration is very low. In order to model this, they should have used a very large supercell.*

The method section of the updated manuscript explicitly mentions that we utilized a 2x2x2 supercell of the primitive spinel structure, which corresponds to a cell with composition $\text{Mg}_{16}\text{Sc}_{32}\text{Se}_{64}$. Several works have been recently published using similar methods and supercell configurations, including *Chem. Mater.*, **27** (17), 6016 (2015) doi: 10.1021/acs.chemmater.5b02342 or Y. Wang *et al.*, *Nature Materials*, **14**, 1026-1031 (2015), doi: 10.1038/nmat4369, or M. Liu *et al.*, *Energy Environ. Sci.*, **8**, 964-974 (2015), doi: 10.1039/C4EE03389B or Richards *et al.* *Nature Communications*, **7**, 11009 (2016), doi: 10.1038/ncomms11009].

In terms of vacancy concentrations within periodic boundary conditions, the referee should note that theoretical models can calculate properties highly accurately as long as the vacancies do not interact with their periodic images significantly. Indeed, periodic cells with high point defect concentrations have been routinely used to accurately calculate defect formation energies and concentrations, as well-documented for semi-conductor applications by Freysoldt *et al.*, “*First-principles calculations for point defects in solids*”, *Rev. Mod. Phys.*, **86**, 253 (2014), doi: 10.1103/RevModPhys.86.253. In the case of theoretical calculations of Mg migration barriers in solids, previous works have shown that the activation barriers do not vary significantly with vacancy concentration in sulfide-spinels [“*A high capacity thiospinel cathode for Mg batteries*”, *Energy Environ. Sci.*, 2016, **9**, 2273-2277, doi: 10.1039/C6EE00724D, and M. Liu *et al.* *Energy Environ. Sci.*, **9**, 3201-3209 (2016), doi: 10.1039/C6EE01731B].

Gerbrand Ceder
Chancellor's Professor of Materials Science and Engineering

References

1. van de Walle, A. & Ceder, G. The effect of lattice vibrations on substitutional alloy thermodynamics. *Rev. Mod. Phys.* **74**, 11–45 (2002).
2. Asta, M., de Fontaine, D., van Schilfgaarde, M., Sluiter, M. & Methfessel, M. First-principles phase-stability study of fcc alloys in the Ti-Al system. *Phys. Rev. B* **46**, 5055–5072 (1992).
3. Ozolins, V., Wolverton, C. & Zunger, A. Cu-Au, Ag-Au, Cu-Ag, and Ni-Au intermetallics: First-principles study of temperature-composition phase diagrams and structures. *Phys. Rev. B* **57**, 6427–6443 (1998).
4. Jain, A. *et al.* Formation enthalpies by mixing GGA and GGA+U calculations. *Phys. Rev. B* **84**, 45115 (2011).
5. Ong, S. P. *et al.* Phase stability, electrochemical stability and ionic conductivity of the $\text{Li}_{10}\pm 1\text{MP}2\text{X}_{12}$ (M = Ge, Si, Sn, Al or P, and X = O, S or Se) family of superionic conductors. *Energy Environ. Sci.* **6**, 148 (2013).
6. Ping Ong, S., Wang, L., Kang, B. & Ceder, G. Li-Fe-P-O₂ Phase Diagram from First Principles Calculations. *Chem. Mater.* **20**, 1798–1807 (2008).
7. Wolverton, C. Crystal structure and stability of complex precipitate phases in Al-Cu-Mg-(Si) and Al-Zn-Mg alloys. *Acta Mater.* **49**, 3129–3142 (2001).
8. Wolverton, C. & Zunger, A. First-Principles Prediction of Vacancy Order-Disorder and Intercalation Battery Voltages in Li_xCoO_2 . *Phys. Rev. Lett.* **81**, 606–609 (1998).
9. Yu, K. & Carter, E. A. Determining and Controlling the Stoichiometry of $\text{Cu}_2\text{ZnSnS}_4$ Photovoltaics: The Physics and

- Its Implications. *Chem. Mater.* **28**, 4415–4420 (2016).
10. Yu, K. & Carter, E. A. Elucidating Structural Disorder and the Effects of Cu Vacancies on the Electronic Properties of $\text{Cu}_2\text{ZnSnS}_4$. *Chem. Mater.* **28**, 864–869 (2016).
 11. Mao, Z., Seidman, D. N. & Wolverton, C. First-principles phase stability, magnetic properties and solubility in aluminum–rare-earth (Al–RE) alloys and compounds. *Acta Mater.* **59**, 3659–3666 (2011).
 12. Opahle, I., Madsen, G. K. H. & Drautz, R. High throughput density functional investigations of the stability, electronic structure and thermoelectric properties of binary silicides. *Phys. Chem. Chem. Phys.* **14**, 16197 (2012).
 13. Reuter, K. & Scheffler, M. Composition, structure, and stability of RuO_2 (110) as a function of oxygen pressure. *Phys. Rev. B* **65**, 35406 (2001).
 14. Johari, G. P. The configurational entropy theory and the heat capacity decrease of orientationally disordered crystals on cooling to 0K. *Philos. Mag. Part B* **81**, 1935–1950 (2001).
 15. Curtarolo, S. *et al.* The high-throughput highway to computational materials design. *Nat. Mater.* **12**, 191–201 (2013).
 16. Curtarolo, S., Morgan, D. & Ceder, G. Accuracy of ab initio methods in predicting the crystal structures of metals: A review of 80 binary alloys. *Calphad* **29**, 163–211 (2005).

REVIEWERS' COMMENTS:

Reviewer #2 (Remarks to the Author):

I think that this work should be published in a more specific journal. I did not see any obvious improvement made by the authors. While they cited many previous results, they did not report the corresponding results to clarify the worries including Referee 3's.

Reviewer #4 (Remarks to the Author):

The manuscript provides interesting theoretical and experimental results. The major concerns of the initial reviewers have been adequately addressed in the revision. After addressing the minor comments below, publication in Nature Communications is recommended.

A more specific title is recommended for this article, which recognizes the focus of the article of spinel structures with AX₂Z₄ chemistries, such as: "High Mg²⁺ Cation Mobility in Solids: An investigation of spinel structures with AX₂Z₄ chemistries".

Based on the discussion in the text and SI, details for construction of the impedance cell are important. Therefore Ta/MgSc₂Se₄/Ta cell for impedance measurement should be described in more detail, including the thickness of the MgSc₂Se₄ pellet, and the Ta coating/foil used for the measurement, and the method of cell assembly.

This review article describing Mg-ion transport and electrochemistry in cathode materials should be added to the introduction section:

- Coordination Chemistry Reviews, 2015, 287, 15–27.

Mg-ion electrochemistry of other spinel structure materials should be cited and discussed in the manuscript, in particular:

- MgMn₂O₄: Chem. Commun., 2017, 53, 3665-3668

Reviewer #5 (Remarks to the Author):

This is a most interesting paper. It reports a joint computational/experimental study of Mg²⁺ migration in solids. This is a key problem in the current field of solid state ionics where there is a strong incentive to develop solid electrolytes based on Mg mobility. As the authors argue, a major problem is the generally low mobility of Mg²⁺ in solids. In this paper, by combining qualitative design considerations together with DFT calculations supported by experiment, the authors demonstrate that high Mg²⁺ mobility can be achieved in a new class of solids.

I have some reservations about the use of GGA-DFT in modelling activated processes in solids, but I note that the work does achieve good agreement between calculations and experiment.

Overall, I consider this to be novel work of high calibre and of general interest which I can recommend for publication in Nature Communications

Subject: Corrections required for NCOMMS-17-08731B

Dear Editor,

Thank you for sending us the referee comments on our manuscript. We are grateful to all the referees for bringing up important points that can improve the overall quality and readability of this novel and original work. Based on the suggestions of the reviewers, changes have been introduced to further strengthen our manuscript. Please find below a detailed response to all the comments and suggestions raised by the reviewers. In general, we appreciate the constructive comments of both reviewers 4 and 5. We hope that the revised manuscript will convince the referee(s) and the editor of the timely importance of this work for the development of materials with high Mg mobility, resulting in a swift publication.

We have adjusted the layout of the manuscript and the supplementary information to comply with the standards of *Nature Communications*.

We would like Nature Communications to include the reviewer reports in the Supporting Information.

Reviewer #4

The manuscript provides interesting theoretical and experimental results. The major concerns of the initial reviewers have been adequately addressed in the revision. After addressing the minor comments below, publication in Nature Communications is recommended.

A more specific title is recommended for this article, which recognizes the focus of the article of spinel structures with AX₂Z₄ chemistries, such as: High Mg²⁺ Cation Mobility in Solids: An investigation of spinel structures with AX₂Z₄ chemistries”.

According to the referee's suggestion and the guidelines provided by Nature Communication we have modified the original title as: **“High Magnesium Mobility in Solids: an Investigation of Ternary Spinel Chalcogenides”**

Based on the discussion in the text and SI, details for construction of the impedance cell are important. Therefore Ta/MgSc₂Se₄/Ta cell for impedance measurement should be described in more detail, including the thickness of the MgSc₂Se₄ pellet, and the Ta coating/foil used for the measurement, and the method of cell assembly.

The revised manuscript now includes a detailed procedure (as below) describing the construction of the Swagelok cell used for the impedance measurements, as well as the thickness information for both electrolyte pellets and the Ta foils that were used as blocking electrodes.

“The resulting pellet, typically with the thickness of 0.5 – 1.0 mm, was assembled into a spring-loaded Swagelok cell, using stainless steel rods as current collectors covered with tantalum foils (~0.05 mm thick) (Sigma Aldrich, ≥99.9%). The preparation of the sample and the Swagelok cell was entirely completed in an Argon glove box.”

This review article describing Mg-ion transport and electrochemistry in cathode materials should be added to the introduction section:

- *Coordination Chemistry Reviews*, 2015, 287, 15–27.

Mg-ion electrochemistry of other spinel structure materials should be cited and discussed in the manuscript, in particular:
- *MgMn₂O₄*: *Chem. Commun.*, 2017,53, 3665-3668

Both articles are now cited and discussed in the introduction section.

Reviewer #5

This is a most interesting paper. It reports a joint computational/experimental study of Mg²⁺ migration in solids. This is a key problem in the current field of solid state ionics where there is a strong incentive to develop solid electrolytes based on Mg mobility. As the authors argue, a major problem is the generally low mobility of Mg²⁺ in solids. In this paper, by combining qualitative design considerations together with DFT calculations supported by experiment, the authors demonstrate that high Mg²⁺ mobility can be achieved in a new class of solids.

I have some reservations about the use of GGA-DFT in modelling activated processes in solids, but I note that the work does achieve good agreement between calculations and experiment.

Overall, I consider this to be novel work of high calibre and of general interest which I can recommend for publication in Nature Communications.

We are thankful for the referee comment.

Reviewer #2

I think that this work should be published in a more specific journal. I did not see any obvious improvement made by the authors. While they cited many previous results, they did not report the corresponding results to clarify the worries including Referee 3's.

We have no further comments on this statement. We have cited existing literature when appropriate to argue the validity of various approaches, rather than re-validate every method in this paper. This is standard in scientific work.

Gerbrand Ceder
Chancellor's Professor of Materials Science and Engineering